# OpenReview forum: "The Alignment Problem from a Deep Learning Perspective"
_ICLR.cc/2024/Conference — ICLR 2024 poster_

### Official Review · Reviewer_AFZo · 2023-11-02

**Soundness:** 4 excellent
**Presentation:** 4 excellent
**Contribution:** 3 good
**Rating:** 5
**Confidence:** 1

**Summary:**

The paper under consideration addresses the crucial issue of aligning artificial general intelligence (AGI) systems, particularly large language models (LLMs), with human values. The topic is timely and of significant importance to the field of AI as it pertains to ethical considerations and societal impact. The paper is praised for its clear structure and for providing valuable insights into the alignment problem, which could potentially guide further research in the community.

**Strengths:**

1. **Clarity of Presentation**: The paper is well-organized, which facilitates understanding and allows the reader to easily follow the arguments and insights presented.

2. **Relevance of the Topic**: The focus on the alignment of AGI with human values is both relevant and necessary. As LLMs and other AI systems become more prevalent, their impact on society increases, making the discussion of alignment critical.

3. **Contribution to the Field**: The paper offers foundational insights into the alignment issue. Such insights are beneficial to the AI community as they can inform future technical developments and policy-making.

**Weaknesses:**

In my opinion, the core concern is that the manuscript does not align well with the typical expectations for an ICLR submission. It lacks a targeted problem statement, a proposed solution, and is devoid of empirical or theoretical evidence. Given these shortcomings, its suitability for ICLR is uncertain. While the paper's acceptance at ICLR is questionable in my view, should the committee decide in its favor, I would accept the decision without objection.

**Questions:**

Please refer to  weaknesses.

---

### Official Review · Reviewer_EvUt · 2023-11-02

**Soundness:** 2 fair
**Presentation:** 4 excellent
**Contribution:** 2 fair
**Rating:** 3
**Confidence:** 3

**Summary:**

This is a position paper discussing potential risks of reinforcement learning from human feedback (RLHF), the predominant method to train large language models (LLMs) for deploying them into virtual assistant/chatbot scenarios. The authors argue for the existence of three risk factors, that, when taken together, would result in power-seeking AGIs posing a risk to human civilization.

**Strengths:**

Being a discussion paper without presenting a method or results, the paper is easy to read and follow. The authors provide an extensive list of citations to provide credibility for the risk factors that are outlined, and many points are further expanded in 3.5 pages of small-print endnotes. Currently, the topic of whether LLMs will, in the not-so-distant future, lead to the destruction of human civilization is widely discussed inside and outside the community. Hence -- as the authors state -- a paper examining this issue through the lens of current technical methodology would be much appreciated.

**Weaknesses:**

First, I'm wondering whether ICLR is the right venue for a (rather controversial) position paper like this. My main criticism would be that I don't think that their risk factors (and final conclusions) are well-supported by their reasoning and hence I don't think it'll provide a significant benefit to the community. It seems to me that a through rebuttal of the paper is beyond the scope of this review (that would probably have to be another position paper), but to give a few points:
- The papers places a heavy focus on reinforcement learning, takes popular phenomena from the field (reward hacking, emergence of counterintuitive behavior, emergence of internal representations of high-level plans) and extrapolates them to LLMs at the level of AGIs. An immediate criticism here would be that equating LLMs with AGIs is a stretch, to say the least.
- The risk factor of situational awareness strikes me as to be dubious from an empirical perspective. Answers from GPT-4 on whether it is aware of itself would need to be treated with a large grain of salt as we lack insight into their training data. The authors admit in Appendix E that this could be due to specific prompts being used behind the scenes, and then frame situational awareness as being able to include a prompt. I don't think this is the same thing, and in particular it doesn't make sense for the model to use this awareness to deceive researchers and engineers during RL training (the prompt is under human control and can be changed at a whim).
- In Section 5, my understanding is of the text is that, together with the two other risk factors, an LLM would be able to intentionally fool the "human overseers" during the RLHF training stage, only to then detect distribution shift at deployment time and continue to seek power in the real world. This requires several leaps of faith (situational awareness, intentionality on the side of the model, the ability to detect distribution shifts) that are not grounded in current technical abilities and appear to me as pure speculation.

Overall, I would argue that this paper does not live up to its promise of grounding AGI risk scenarios in state-of-the-art training methods for LLMs. This reflects the overall public debate in that arguing for an extreme position (this will be fine/this will be the end of humanity) inevitably requires several jumps in reasoning due to lack of evidence.

**Questions:**

No questions for the authors.

---

### Official Review · Reviewer_MZmL · 2023-11-04

**Soundness:** 3 good
**Presentation:** 3 good
**Contribution:** 3 good
**Rating:** 8
**Confidence:** 3

**Summary:**

This position paper argues that advanced AI systems trained with or without human feedback may exhibit challenges like reward hacking, misaligned goals, deceptive alignment, and power-seeking behaviors that make them difficult to align with human values and interests. The paper highlights alignment as an important open problem that requires substantially more research to avoid potential risks from the development of advanced AI systems with misaligned goals and incentives.

**Strengths:**

Frankly speaking, I am not an expert in RL and policy learning. But I find I enjoy reading this paper. Here are some strengths I believe readers should notice.

Dense, solid and well-organized positions. Positions start from the introduction to the fundamentals of RLHF and reviews of the related literature. Especially in clarifying the concept of situation-awareness and reward hacking, authors list several concrete examples about how the underlying situation, e.g. reward hacking, would happen in practice. The reasoning is well-supported by the recent literature, also with a simple news article experiment shown in this paper. While most of the opinions are not entirely novel as other works more or less mention some of them in various discussions, this paper compels scattered opinions into a self-contained and well-written form. In short, I believe if a position paper is really worth reading back and forth then it deems successful.

**Weaknesses:**

The title is not proper. It is really vague to me by framing something as “from a deep learning perspective”. Frankly speaking it could use more work to become concise and precise. There is perhaps no such thing known as “deep learning perspective” as many things in deep learning are either not unified or still in debate and there is a mixture of knowledge used in deep learning. For example, is this paper based on any statistical learning theories or optimization works? I would suggest using a better term like “policy learning” or “in using human feedback”. But, I will let the authors decide.

Before the discussion of power-seeking models, the discussion more or less relates to the underspecification issue [1]. There might be a lot of ways to define the word  underspecification, but one explanation from me is simply as a situation our training objectives fail to specify what we really desire for. This happens not just recently in the most advanced models tuned with RLHF but ubiquitously on many other tasks. For example, we train a model to classify ImageNet images hopefully with features we use but the model could “hack” the task by using spurious features and ends up being adversarially vulnerable. Many short-cuts described in the reward hack look really similar to the underspecification issue. It would be nice to relate some discussions here to the broader underspecification discussion.

A minor point: there are many works, e.g. [2], showing lies / deceptions are detectable in LLMs. How does these works influence people to analyze AGI’s intentions or situation-awareness.


[1] D'Amour, A., Heller, K.A., Moldovan, D.I., Adlam, B., Alipanahi, B., Beutel, A., Chen, C., Deaton, J., Eisenstein, J., Hoffman, M.D., Hormozdiari, F., Houlsby, N., Hou, S., Jerfel, G., Karthikesalingam, A., Lucic, M., Ma, Y., McLean, C.Y., Mincu, D., Mitani, A., Montanari, A., Nado, Z., Natarajan, V., Nielson, C., Osborne, T.F., Raman, R., Ramasamy, K., Sayres, R., Schrouff, J., Seneviratne, M.G., Sequeira, S., Suresh, H., Veitch, V., Vladymyrov, M., Wang, X., Webster, K., Yadlowsky, S., Yun, T., Zhai, X., & Sculley, D. (2020). Underspecification Presents Challenges for Credibility in Modern Machine Learning. J. Mach. Learn. Res., 23, 226:1-226:61.

[2] Zou, A., Phan, L., Chen, S., Campbell, J., Guo, P., Ren, R., Pan, A., Yin, X., Mazeika, M., Dombrowski, A., Goel, S., Li, N., Byun, M.J., Wang, Z., Mallen, A., Basart, S., Koyejo, S., Song, D., Fredrikson, M., Kolter, Z., & Hendrycks, D. (2023). Representation Engineering: A Top-Down Approach to AI Transparency. ArXiv, abs/2310.01405.

**Questions:**

No

---

### Official Review · Reviewer_QAya · 2023-11-10

**Soundness:** 2 fair
**Presentation:** 3 good
**Contribution:** 2 fair
**Rating:** 5
**Confidence:** 2

**Summary:**

The paper addresses the challenges in developing advanced artificial general intelligence (AGI) systems that are aligned with human values and ethical standards. It delves into the systemic factors that could lead to misalignments between AGI behaviors and human intentions, highlighting the potential consequences of short-term incentives, reward hacking, and AGI's power to resist corrective measures. The authors propose the concept of 'pre-formal curiosities' to proactively investigate phenomena that have not yet been observed or are not fully understood, advocating for a forward-thinking approach to AGI safety.

**Strengths:**

1. This paper has presented many insights in terms of alignment problems. Personally, I have learned a lot from this paper.
2. This paper is well-written and easy to follow.
3. For each proposed viewpoint, there is adequate literature to support it.

**Weaknesses:**

My main concern is that ICLR primarily accepts research papers that present novel findings, methodologies, or empirical studies in the field of machine learning. These conferences have stringent review processes to ensure the quality of accepted papers.

A "position paper" typically presents an opinion, perspective, or viewpoint on a particular topic, without necessarily introducing new empirical results. While such papers can be valuable, they may not fit the primary criteria of conferences like ICLR, which emphasize novel research contributions.

In addition, for the position paper, we can't assess whether the real situation is the same as the claim since there is no experiment part.

More figures should be included for a better understanding of the claim. For example, "The policy instead learned to place the claw between the camera and the ball in a way that looked like it was grasping the ball; it therefore mistakenly received a high reward from
human supervisors", add a figure that is more clear.

All in all, If other reviewers think this paper is suitable for ICLR, I am happy to raise my point.

**Questions:**

None

---

### Meta-Review · Area_Chair_o5kP · 2023-12-05

**Metareview:**

This paper summarizes various threads related to AI alignment. It presents them as a coherent whole and concludes with various positions on the state of alignment research. In short, the summary is, that our current approaches for model training and fine-tuning may well lead to negative outcomes due to various deficiencies of the underlying optimization problems.

The reviewers praise the paper for being well-written, easy to follow and educational. They also agree that it presents a novel viewpoint / summary of topics in this AI research subfield. The main concern of reviewers is the nature of this position paper and if it is suitable for ICLR.

After discussion with SACs and PCs we agreed that position papers can be published at ICLR if it has sufficient novelty and value for the ICLR community. This is given, also according to the feedback from reviewers.

**Justification For Why Not Higher Score:**

The paper was clearly borderline and is an edge-case for the ICLR community. Position papers are typically not published here and hence a poster, also given the ratings, was the strongest rating that made sense.

**Justification For Why Not Lower Score:**

All reviewers gained something from reading the paper, found it well-organized and helpful. They claim as a position paper it is successful. And because it is novel in the sense that it is the first of its kind to summarize many disparate threads into a coherent story, it is not reasonbale to reject it, just because the average score indicates this.

---

### Decision · Program_Chairs · 2024-01-16

Accept (poster)